# Meta-Analysis of Survival Effects of Receptor Tyrosine Kinase-like Orphan Receptor 1 (ROR1)

**DOI:** 10.3390/medicina58121867

**Published:** 2022-12-17

**Authors:** Soo Young Jeong, Kyung-jun Lee, Jieum Cha, So Yoon Park, Hyeong Su Kim, Jung Han Kim, Jae-Jun Lee, Namhyeok Kim, Sung Taek Park

**Affiliations:** 1Department of Obstetrics and Gynecology, Kangnam Sacred-Heart Hospital, Hallym University Medical Center, Hallym University College of Medicine, Seoul 07441, Republic of Korea; 2Institute of New Frontier Research Team, Hallym University, Chuncheon 24252, Republic of Korea; 3Division of Hemato-Oncology, Department of Internal Medicine, Kangnam Sacred-Heart Hospital, Hallym University Medical Center, Hallym University College of Medicine, Seoul 07441, Republic of Korea; 4Departments of Anesthesiology and Pain Medicine, Chuncheon Sacred-Heart Hospital, Hallym University Medical Center, Hallym University College of Medicine, Chuncheon 24253, Republic of Korea

**Keywords:** cancer, meta-analysis, ROR1, receptor tyrosine kinase-like orphan receptor 1

## Abstract

*Background and Objectives*: Identification and targeting of membrane proteins in tumor cells is one of the key steps in the development of cancer drugs. The receptor tyrosine kinase-like orphan receptor (ROR) type 1 is a type-I transmembrane protein expressed in various cancer tissues, which is in contrast to its limited expression in normal tissues. These characteristics make ROR1 a candidate target for cancer treatment. This study aimed to identify the prognostic value of ROR1 expression in cancers. *Materials and Methods*: We conducted a comprehensive systematic search of electronic databases (PubMed) from their inception to September 2021. The included studies assessed the effect of ROR1 on overall survival (OS) and progression-free survival (PFS). Hazard ratios (HR) from collected data were pooled in a meta-analysis using Revman version 5.4 with generic inverse-variance and random effects modeling. *Results*: A total of fourteen studies were included in the final analysis. ROR1 was associated with worse OS (HR 1.95, 95% confidence interval (CI) 1.50–2.54; *p* < 0.001) with heterogeneity. The association between poor OS and ROR1 expression was high in endometrial cancer, followed by ovarian cancer, and diffuse large B cell lymphoma. In addition, ROR1 was associated with poor PFS (HR 1.84, 95% CI 1.60–2.10; *p* < 0.001), but heterogeneity was not statistically significant. In subgroup analysis, high ROR1 expression showed a significantly higher rate of advanced stage or lymph node metastasis. *Conclusions*: This meta-analysis provides evidence that ROR1 expression is associated with adverse outcome in cancer survival. This result highlights ROR1 as a target for developmental therapeutics in cancers.

## 1. Introduction

Patients with cancer have been treated with surgery, chemotherapy, and radiotherapy. In recent decades, individualized approaches targeting gene alterations or membrane proteins on tumor cells have emerged as novel therapies for patients with cancer [1]. Therefore, the challenge of identifying novel biomarkers or surface proteins continues, and new strategies, such as antibody–drug conjugates (ADCs) or adaptive cell therapy, have been introduced into clinical practice [2,3].

Receptor tyrosine kinase-like orphan receptor type 1 (ROR1), a type I orphan receptor tyrosine kinase-like surface protein, is a key protein in embryogenesis and its expression is restricted in normal tissues. The Wnt signaling pathway is a developmental pathway in embryogenesis and regulates cell proliferation and migration during organogenesis through two distinct arms: β-catenin-dependent (canonical) and β-catenin-independent (non-canonical) [4]. It is known that dysregulation of these pathways has been identified in several cancers [5,6]. β-Catenin-dependent Wnt signaling has been associated with breast, colon, gastric, and endometrial cancer development by mutations in β-catenin or adenomatous polyposis coli (APC) [6,7,8]. Additionally, binding of Wnt5a ligand to ROR1 receptor can activate β-catenin-independent Wnt signaling and result in cell proliferation and migration related with carcinogenesis [9,10,11].

However, several studies have shown that ROR1 is highly expressed in neoplastic cells in many different types of cancers, such as leukemia, breast cancer, and ovarian cancer [12,13,14,15,16,17]. It is also associated with the expression of several epithelial–mesenchymal transition markers [16,18], tumor cell proliferation, and relapse [19,20]. Some studies have reported that ROR1 expression is a potential predictive factor for poor survival [21,22]. As such, this surface antigen has been considered as a new target candidate for cancer treatment.

Several preclinical studies evaluating monoclonal antibodies [23,24] or chimeric antigen receptor (CAR)-T cells [25] in leukemia have been conducted. The present meta-analysis assessed whether ROR1 expression has an impact on survival in various types of cancer using published data.

## 2. Materials and Methods

### 2.1. Data Sources and Searches

The review was conducted in accordance with the Preferred Reporting Items for Systematic Reviews and Meta-analyses (PRISMA) guidelines [26] and recommendations of the Cochrane Collaboration [27]. We performed electronic searches of MEDLINE (host: PubMed) from 1946 to September 2021 using keywords such as “receptor tyrosine-kinase-like orphan receptor 1” or “ROR1” for intervention; “cancer” or “malignancy” for disease; and “progression”, “survival” or “recurrence” for outcome, and the total number of searched literatures was 243.

### 2.2. Study Selection

Studies that met the following conditions were included in this meta-analysis: (1) patients had been diagnosed with hematologic or solid cancer; (2) studies with comparative results according to ROR1 expression level; (3) studies reporting a hazard ratio (HR) for progression-free survival (PFS), overall survival (OS), or survival curves, which allowed for the estimation of HR for survivals; and (4) English language publications. Case reports, conference abstracts, or letters were excluded from the study. If a HR with a 95% confidence interval (CI) for survival was not reported directly and could not be calculated, the study was also removed.

### 2.3. Data Extraction

The following variables were collected for coding the study: publication year, name of first author, journal, country, type of cancer, number of patients included in the analysis, detection methods, agents used for ROR1 expression, cutoff values used for ROR1 intensity, and cancer status (stage, lymph node (LN) metastasis). The primary outcome was OS and the secondary outcome was PFS according to ROR1 expression. In some primary reports, these parameters were extracted directly. In other studies, we estimated them from other reported survival curves using the methods described by Parmar et al. [28]. Additionally, univariable HR and estimated HRs from survival plots were collected using a hierarchal approach.

### 2.4. Statistical Analysis

The analyses were performed using ReyMan version 5.4 analysis software (Cochrane Collaboration, Copenhagen, Denmark). Extracted HRs were pooled and weighted by generic inverse variance and computed by random effects modeling. Statistical heterogeneity in the results of the studies was assessed by Cochran’s Q and expressed as the I^2^ index [29]. Subgroup analyses were performed for cancer types, stages, and status of LN metastasis. A funnel plot was produced to assess the possibility of publication bias. Two-sided tests were applied and *p*-values < 0.05 were considered statistically significant.

## 3. Results

### 3.1. Literature Search and Reporting of Information

A flowchart of the systematic search process is summarized in Figure 1. A total of 243 retrospective studies were identified in the MEDLINE database (host: PubMed). After review by all the authors, a total of 14 studies, including 4035 subjects, qualified for this study [16,21,22,30,31,32,33,34,35,36,37,38,39,40]. Detailed descriptions of all included studies are outlined in Table 1. Of the 14 studies, three were conducted on leukemia, three on breast cancer, two on ovarian cancer, two on endometrial cancer, two on lung cancer, one on gastric cancer, and one on colorectal cancer. The funnel plot was symmetrical in appearance (Figure 2).

### 3.2. Survival

In all 14 studies, ROR1 was significantly associated with poor OS (HR 1.95, 95% CI 1.50–2.54; *p* < 0.001, Figure 3A). The sample was heterogeneous (χ2 = 48.14, df = 13, Cochran’s Q *p* < 0.001, I^2^ = 73%).

For subgroup analysis for each cancer type, endometrial cancer (HR 3.07, 95% CI 1.62–5.79, *p* = 0.0006), ovarian cancer (HR 2.37, 95% CI 1.18–4.79, *p* = 0.02), B-cell lymphoma/leukemia (HR 2.03, 95% CI 1.53–2.69, *p* < 0.0001), and breast cancer (HR 1.73, 95% CI 1.19–2.54, *p* = 0.004) showed significant results indicating better prognosis of low ROR1 expression.

Data on the association between ROR1 and PFS were reported in five studies, including triple-negative breast cancer, chronic lymphocytic leukemia (CLL), and gynecologic cancers. ROR1 was associated with poor PFS (HR 1.84, 95% CI 1.60–2.10; *p* < 0.001, Figure 3B). Heterogeneity was not observed (χ2 = 5.84, df = 4, Cochran’s Q *p* = 0.21, I^2^ = 31%) because there were few studies which reported PFS. Triple negative breast cancer had significantly worse PFS (HR 3.06, 95% CI 1.36–6.85), followed by ovarian cancer (HR 2.90, 95% CI 1.64–5.13) and endometrial cancer (HR 2.45, 95% CI 1.21–4.97).

Four studies were included in the analysis of odds ratio (OR) for tumor stage (I/II vs. III/IV) and LN metastasis (Appendix A) [33,35,37,39]. There was a significant association between ROR1 expression and advanced stages (OR 4.60, 95% CI 2.05–10.31, *p* = 0.0002) (Figure 4A), and tumors with high ROR1 expression showed a significantly higher rate of LN metastasis (OR 8.27, 95% CI 4.17–16.39, *p* < 0.00001) (Figure 4B).

## 4. Discussion

The receptor tyrosine kinase ROR1, a transmembrane glycoprotein, plays an important role in embryogenesis and is overexpressed in many malignant tumors, such as leukemia [12,15], breast cancer [17,18], prostate cancer, and ovarian cancer [22]. Several studies have shown that it plays a critical role in carcinogenesis by activating cell survival signaling events with non-canonical Wnt signaling pathways [41]. Interestingly, ROR1 was shown to be enriched in chemoresistant breast and ovarian cancers. Fultang et al. recently reported that ROR1 is an upstream regulator of the adenosine triphosphate (ATP)-binding cassette (ABC) transporter protein (ABCB1), and promotes its stability in breast cancer [42]. Hanna et al. reported that ROR1-mediated stemness promotes chemoresistance in ovarian cancer [43]. This study revealed that ROR1 expression was significantly associated with poor prognosis in various cancer types. Given the aberrant expression of ROR1 in cancer cells and its important role in cell proliferation, new therapeutic strategies targeting ROR1 have been evaluated in preclinical studies and clinical trials. Among these, monoclonal antibody (mAbs)-based strategies have advanced the most [24,44,45]. The mAbs can bind ligands directly and induce antibody-dependent cellular cytotoxicity or complement–dependent cytotoxicity [46]. Cirmtuzumab, the first humanized mAb targeting ROR1, inhibits ROR1 signaling and stemness signatures in CLL [25]. Clinical trials evaluating the efficacy of its combination with other agents are ongoing: a phase Ib/II trial of cirmtuzumab and Ibrutinib in B cell CLL and mantle cell lymphoma (NCT03088878, NCT03420183), and a phase Ib trial of cirmtuzumab and paclitaxel in breast cancer (NCT02776917, NCT02860676).

Other strategies, such as ADC, bispecific T-cell engager, and CAR T-cells, have also been developed. VLS-101 (a combination of mAb and monomethyl auristatin E) and NBE-002 (a combination of mAb and PNU-159682) are ADCs that have been demonstrated to have anti-cancer effects in preclinical studies [47]. They have also been evaluated in clinical trials (NCT03833180, NCT04504916, NCT04441099). NVG-111, a bispecific antibody targeting ROR1 and CD3, showed anti-tumor effects by T-cell mediated cytotoxicity in preclinical data [48]. ROR1-CAR T-cells are cytotoxic specific to ROR1-expressed tumor cells [49,50]. A phase I trial with ROR1-CART-cells in refractory hematologic malignancies, breast cancer, and lung cancer is ongoing (NCT02706392).

This study has some limitations. First, all included studies were retrospective; therefore, they might have been affected by publication bias and several methodological shortcomings. In addition, there was variation in effect sizes (heterogeneity), scoring systems, and antibodies among the studies. Furthermore, we calculated HR estimates from survival plots in several studies that did not report HRs. This could have resulted in inaccurate estimates.

## 5. Conclusions

This meta-analysis revealed that ROR1 expression is associated with poor survival in patients with various cancers. Our results suggest that ROR1 expression may be a valuable prognostic biomarker for identifying patients at higher risk of mortality. Considering that ROR1 is a membrane protein and an excellent target for immunotherapy, there is urgent need for new therapeutic agents that target ROR1.

## Figures and Tables

**Figure 1 medicina-58-01867-f001:**
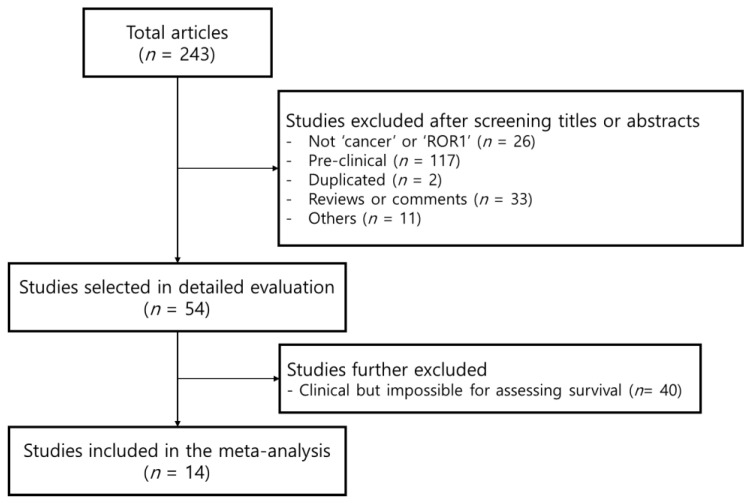
Flowchart for the receptor tyrosine kinase-like orphan receptor 1 (ROR1) data selection.

**Figure 2 medicina-58-01867-f002:**
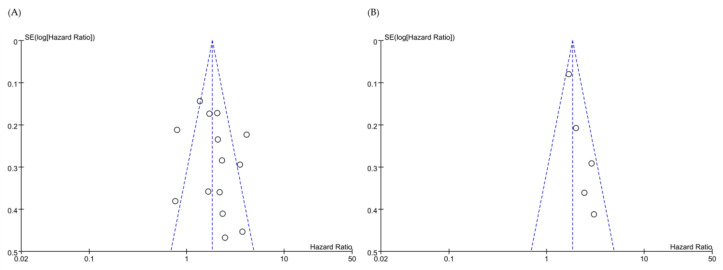
Funnel plot for (**A**) overall survival, and (**B**) progression-free survival, according to ROR1 expression. Dots(o) represent individual studies.

**Figure 3 medicina-58-01867-f003:**
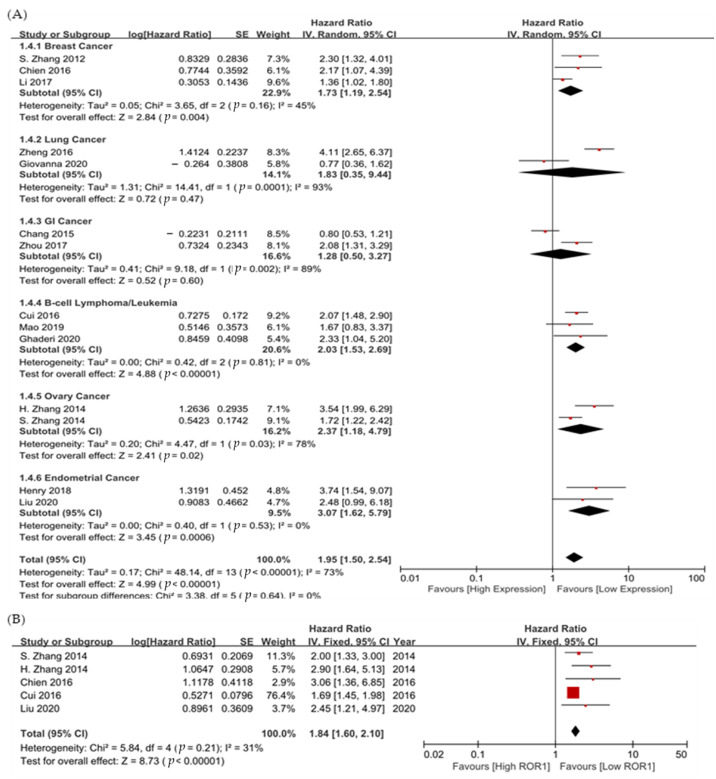
Forest plot showing hazard ratios for (**A**) overall survival, and (**B**) progression-free survival, according to ROR1 expression.

**Figure 4 medicina-58-01867-f004:**
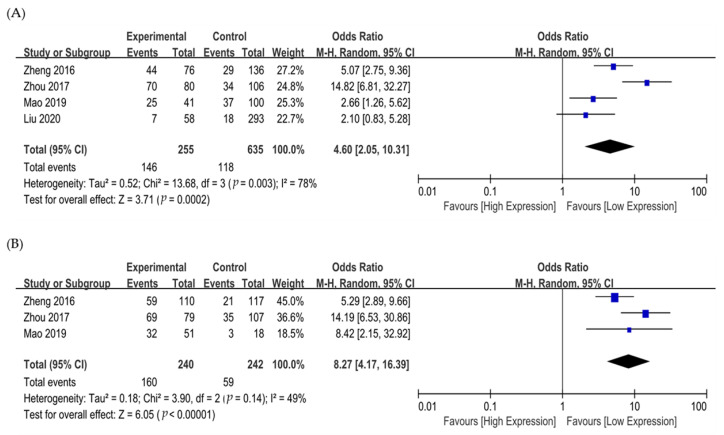
Forest plot of odds ratio for (**A**) advanced stages (I/II vs. III/IV) and (**B**) lymph node metastasis.

**Table 1 medicina-58-01867-t001:** Characteristics included for the study of ROR1.

First Author (Year) [Ref]	Country	Cancer Type	No.	Detection Method	Agent Used	Cutoff Used	HR for PFS (95% CI)	HR for OS (95% CI)
S. Zhang (2012) [16]	China	Breast cancer	295	IHC	Alexa-647-conjugated monoclonal Ab (mAb; 4A5)	Intensity score > 50%	-	2.30 (1.32–4.01) *p* = 0.003
H. Zhang (2014) [30]	China	Ovarian cancer	100	IHC	Rabbit polyclonal Ab (1:200, Abcam)	Staining score * ≥ 2	2.90 (1.64–5.13) *p* = 0.01	3.54 (1.99–6.29) *p* = 0.01
S. Zhang (2014) [22]	U.S.A.	Ovarian cancer	285	RT-PCR	N/A	Upper third expression of ROR1 mRNA	2.0 (1.4–3.0) *p* = 0.0003	1.72 (1.22–2.42) *p* < 0.05
H. Chang (2015) [31]	Republic of Korea	Gastric cancer	424	IHC	Rabbit polyclonal antibody (1:25; Abcam)	Staining > 50%	-	0.8 (0.53–1.21) *p* = 0.189
Chien (2016) [21]	China	Triple negative breast cancer	210	IHC	Rabbit polyclonal antibody (1:100 dilution; proteintech)	Staining score * ≥ 2	3.06 (1.36–6.86) *p* = 0.007	2.17 (1.07–4.39) *p* = 0.031
Cui (2016) [32]	U.S.A.	CLL	1568	IHC	Alexa-647-conjugated monoclonal Ab (mAb; 4A5)	ΔMFI (mean fluorescence intensity) > 32	1.69 (1.45–1.98) *p* < 0.0001	2.07 (1.48–2.90) *p* < 0.0001
Zheng (2016) [33]	China	Lung cancer	161	IHC	Anti-ROR1 (Abcam 135669, 1:20)	Staining score * > 2	-	4.11 (2.51–6.38) *p* < 0.001
Zhou (2017) [35]	China	Colorectal	186	IHC	Polyclonal rabbit Ab (1:20, Abcam)	Staining score * > 2	-	2.08 (1.31–3.29) *p* = 0.002
Li (2017) [34]	U.S.A.	Triple negative breast cancer	150	IHC	N/A	N/A	-	1.357 (1.024–1.798) *p* = 0.0336
Henry (2018) [36]	Australia	Endometrial cancer	87	IHC	Anti-ROR1 (Abcam ab135669)	Intensity score = 3	-	3.74 (1.54–9.07) *p* = 0.004
Mao (2019) [37]	China	DLBCL	150	IHC	Primary polyclonal rabbit anti-ROR1 antibody (ab135669, 1:300, Abcam, Cambridge, MA)	Staining score * ≥ 4	-	1.67 (0.831–3.370), *p* = 0.149
Liu (2020) [39]	Australia	Endometrial cancer	330	IHC	Anti-ROR1 (1:50, #564464, BD Bioscience)	Intensity score = 3	2.45 (1.21–4.97) *p* = 0.01	2.48 (0.99–6.18) *p* = 0.05
Giovanna (2020) [40]	Switzerland	Lung adenocarcinoma	56	qRT-PCR		>median of expression	-	0.769 (0.364–1.62) *p* = 0.4915
Ghaderi (2020) [38]	Sweden	DLBCL	33	IHC	Polyclonal antibody against ROR1 (Proteintech, Manchester, United Kingdom)	>10% (level of unequivocal cytoplasmic and/or membranous staining in the neoplastic B cells)	-	2.33 (1.04–5.20) *p* = 0.039

* Staining score = product of staining intensity (0–3) and percentage of ROR1 positive (1–4). ROR1, receptor tyrosine kinase-like orphan receptor 1; HR, hazard ratio; IHC, immunohistochemistry; RT-PCR, reverse transcription-polymerase chain reaction; PFS, progression-free survival; OS, overall survival; CLL, chronic lymphocytic leukemia; DLBCL, diffuse large B cell lymphoma.

## Data Availability

Not applicable.

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
