# Peer review of "Meta-Analysis of Survival Effects of Receptor Tyrosine Kinase-like Orphan Receptor 1 (ROR1)"

_medicina, 2022, doi:10.3390/medicina58121867_

Round 1

Reviewer 1 Report

In this article, the authors determined the clinical significance of ROR in various cancer by meta-analysis, including 14 studies after selection. They have mentioned the criteria of inclusion and exclusion. They have mentioned the information of each of the studies. Regarding the analysis, it is too general. Overall, they claimed that high expression of ROR would be associated with poor survival. However, more specific information will be required, especially the disease stage, which they did not include in the analysis. A major revision will be required.

Specific point:

1.       Since cancer has different subtypes, e.g. ER+ve, HER2 and TNBC for breast cancer, small cell and non-small cells for lung cancer, etc. The authors did not examine if ROR would be associated with a particular subtype of each cancer. Therefore, the study did not make any advancement.

2.       How about the expression fo ROR in different tumor stages?

3.       Regarding the survival analysis, how about the survival curve of each cancer?

4.       There are many biomarkers associated with poor survival, and they are used for guiding the treatment. For example, endocrine therapy will be used for ER+ve breast cancer. The authors have mentioned clinical trials examining the effect of targeting ROR. The authors would need to discuss how the researchers defined ROR+ve. Did it match the data from the survival analysis?  

Reviewer 2 Report

1. Line 42. Typographical error "However, However, several studies ...."

2. Add more information about Receptor tyrosine kinase-like orphan receptor type 1 (ROR1), in the introduction section.

3. Line 60. How many literatures (no. of literature) you have downloaded from database.

4. Line 72. I think it should be mentioned in Author's contribution section and the sentence "Two authors (SYP and JEC) independently reviewed and extracted the data from all 71 included studies and one reviewer (STP) was consulted to resolve disagreement. " should be rephrased.

5. List of abbreviation should be added in the manuscript.

Round 2

Reviewer 1 Report

The authors have already addressed all the concerns.

Reviewer 2 Report

Manuscript is ok now.